# The Large American Liver Fluke (*Fascioloides magna*): A Survivor's Journey through a Constantly Changing World

Ágnes Csivincsik [1,*], Tibor Halász [2] and Gábor Nagy [1]

1   One Health Working Group, Institute of Physiology and Animal Health, Hungarian University of Agriculture and Life Sciences, 7400 Kaposvár, Hungary; nagy.gabor.oh@uni-mate.hu
2   Zselic Wildlife Estate, SEFAG LLC, 7400 Kaposvár, Hungary; halasz.tibor@sefag.hu
*   Correspondence: csivincsik.agnes@uni-mate.hu

**Abstract:** The large American liver fluke (*Fascioloides magna*) is considered an invasive trematode parasite in Europe. Its origin dates back before the Cretaceous-Paleogene Mass Extinction, after which it survived at least three population bottlenecks and successful host switches before it arrived in Europe. The authors review the evolutionary history of *F. magna*, the distribution by its ancient proboscidean hosts, and the probable drivers of the switch to the white-tailed deer (*Odocoileus virginianus*). The review collects knowledge on the biology of intermediate hosts, which helps understand the factors that influence the epidemiology of *F. magna* in aquatic ecosystems. The authors demonstrate the adaptation potential of the parasite using data from both North American and European endemics. Finally, the study calls attention to the epidemiological risk of human-induced global change, with a special interest in the invasive snail species *Pseudosuccinea columella*.

**Keywords:** *Fascioloides magna*; Proboscidea; host switch; bottleneck; virulence; *Pseudosuccinea columella*

## 1. Introduction

The large American liver fluke (Fascioloides magna) belongs to the Digenean subclass of the Trematode class. *Protofasciola robusta*, *Fasciolopsis buski*, *Fasciola hepatica*, *F. gigantica*, *F. nyanzae*, *Tenuifasciola tragelaphi*, and *Fascioloides jacksoni* are members of the Fasciolidae family (Figure 1) [1]. This taxon consists of several endoparasites of human and animal health importance. Their development is indirect, which means their life cycle needs an intermediate host. The fluke possesses both female and male genitals; they are hermaphrodites [2,3] *F. magna* can parasitize mostly cervids, less frequently other ruminants, and sometimes suids, horses, and rodents; contrary to *Fasciola hepatica*, *F. gigantica*, and *F. buski*, the zoonotic relevance of *F. magna* has not been proved [2].

The first description of the species was published in 1875 by Bassi. That work was based on the discovery of an isolated population of the parasite in the Regional Park La Mandria near Turin, Italy [3–5]. In this work, the parasite was described as very similar to *F. hepatica* in view of both its morphology and the clinical signs it caused. At its first discovery, the parasite was named *Distomum magnum* [4]. After this first European detection, several authors described the species during the last decades of the 19th century on the North American continent, and its American origin was confirmed [5–7].

In final hosts, by causing mild to severe lesions of the liver parenchyma, *F. magna* affects hepatic function, thus the metabolism of the body and the performance of the host [8–10]. The parasite reaches maturity and produces eggs in the body of the final host. The eggs of the parasite are shed by the faeces of the hosts. In the environment, protected by the humid and warm conditions of the faecal material, the eggs begin to develop and hatch after 35 days. If the temperature is lower than its optimum (24–28 °C), this process gets longer. From the eggs, free-swimming miracidia hatch and begin to seek intermediate hosts to continue their development [6]. In their life cycle, amphibian snails of the Lymnaeidae

family play the role of the intermediate host [7,11,12]. Therefore, the distribution of the parasite depends on the freshwater habitats where its intermediate hosts live (Figure 2) [13].

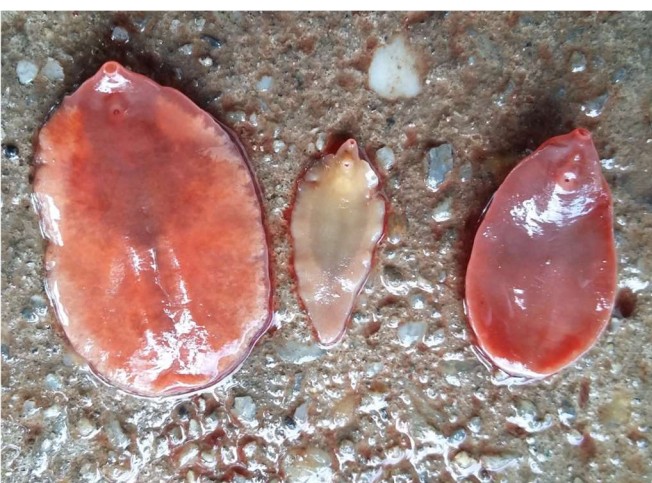

**Figure 1.** European representatives of the Fasciolidae family: *Fasciola hepatica* (in the middle) and *Fascioloides magna*. In addition, the size, "shoulders" differentiate *F. hepatica* from *F. magna*.

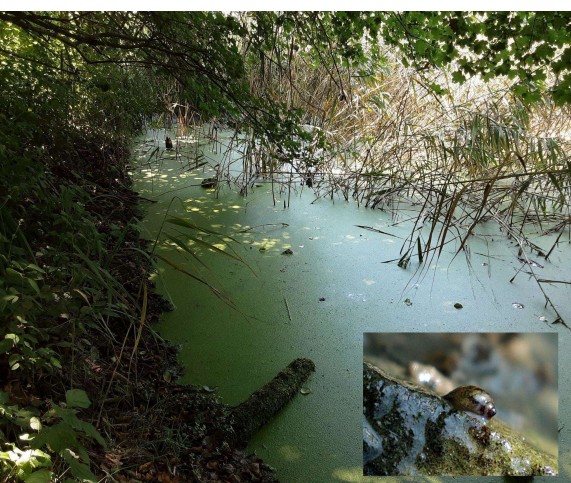

**Figure 2.** Lentic freshwater ecosystem with shallow shores and rich aquatic plant community, which provide ideal habitat for lymnaeid snails (small picture).

In the intermediate hosts' parasitic developmental stage, redia, the first mother redia, and then the daughter rediae will evolve. Daughter rediae produce cercariae, the next free-swimming developmental stage of *F. magna*. Cercariae escape from the body of the snail nocturnally. Swimming cercariae search for a shady environment, mostly the surface of aquatic plants, on which they go through encystation, and form metacercariae. Metacercaria is a circular-shaped developmental stage that is glued to aquatic plants by adhesive substance coverage. In this form, they can survive for longer periods, even in partial desiccation, waiting for the final hosts to uptake them during grazing on infested plants (Figure 3) [6].

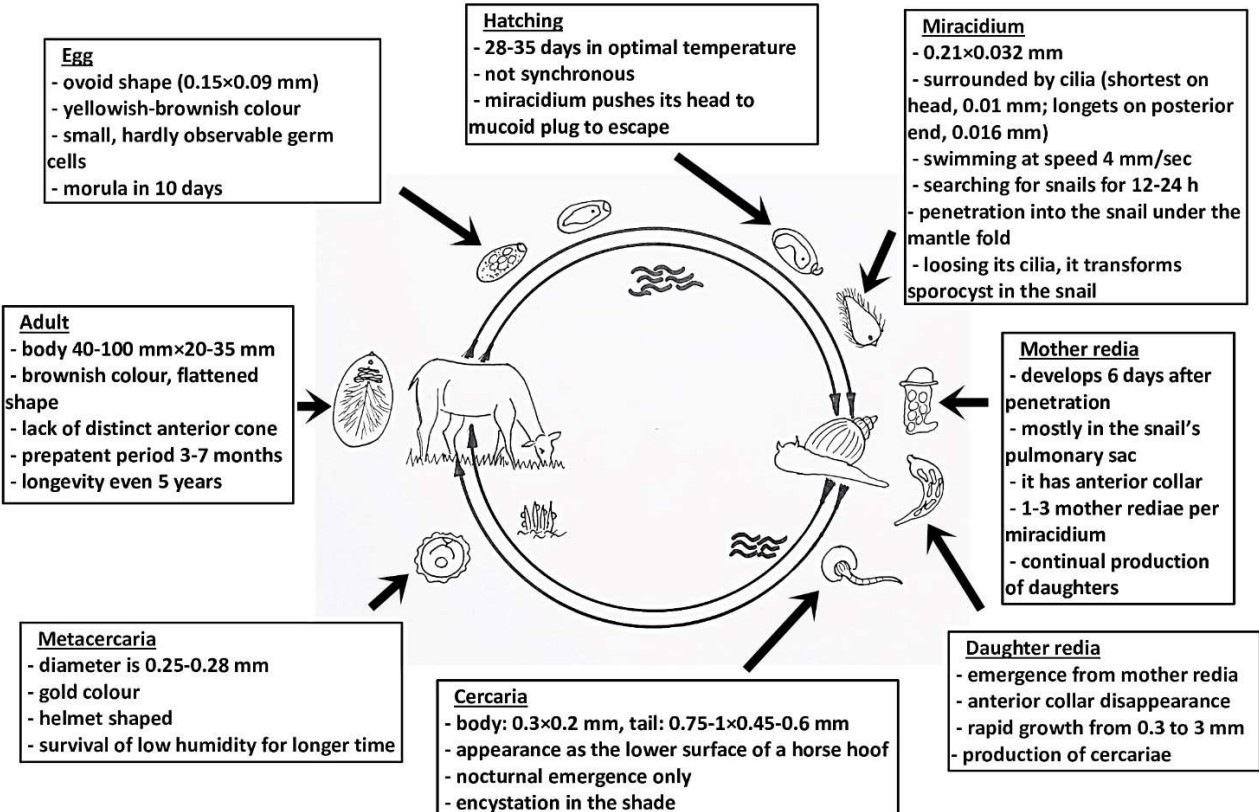

**Egg**
- ovoid shape (0.15×0.09 mm)
- yellowish-brownish colour
- small, hardly observable germ cells
- morula in 10 days

**Hatching**
- 28-35 days in optimal temperature
- not synchronous
- miracidium pushes its head to mucoid plug to escape

**Miracidium**
- 0.21×0.032 mm
- surrounded by cilia (shortest on head, 0.01 mm; longets on posterior end, 0.016 mm)
- swimming at speed 4 mm/sec
- searching for snails for 12-24 h
- penetration into the snail under the mantle fold
- loosing its cilia, it transforms sporocyst in the snail

**Adult**
- body 40-100 mm×20-35 mm
- brownish colour, flattened shape
- lack of distinct anterior cone
- prepatent period 3-7 months
- longevity even 5 years

**Mother redia**
- develops 6 days after penetration
- mostly in the snail's pulmonary sac
- it has anterior collar
- 1-3 mother rediae per miracidium
- continual production of daughters

**Metacercaria**
- diameter is 0.25-0.28 mm
- gold colour
- helmet shaped
- survival of low humidity for longer time

**Cercaria**
- body: 0.3×0.2 mm, tail: 0.75-1×0.45-0.6 mm
- appearance as the lower surface of a horse hoof
- nocturnal emergence only
- encystation in the shade

**Daughter redia**
- emergence from mother redia
- anterior collar disappearance
- rapid growth from 0.3 to 3 mm
- production of cercariae

**Figure 3.** Life cycle of the large American liver fluke (*Fascioloides magna*): designed by Á. Csivincsik based on the works of Kassai [2], Swales [6], and Campbell [14].

At present, *F. magna* occupies natural habitats in North and Central America, Europe, South Africa, and Australia [2,3,5]. From its original North American habitats, it travelled to other continents by transporting infected hosts (mostly cervids). The history, especially the European history, of the large American liver fluke demonstrates the central role of human activities in the spread of this exotic parasite. For now, *F. magna* has become an important element of global change [15]. Our review attempts to demonstrate the evolutionary, epidemiological, and pathological features that contributed to its successful distribution.

## 2. Evolutionary History of the Large American Liver Fluke and Its Relatives

### 2.1. The Cretaceous-Paleogene Boundary

Little evidence is available on the distribution and host specificity of the early Digenean trematodes [16]. It is hypothesised that, before adaptation to vertebrate hosts, these trematodes parasitised molluscs [13]. Some excavations revealed that Digenean trematodes can be found in dinosaur coprolites and mollusc fossils from the Upper (Late) [17] and Lower (Early) [18,19] Cretaceus periods; nevertheless, Cretaceous-Paleogene Mass Extinction (KPgME) around 66 million years ago (MYA) swept away most host species with their parasites [16].

The most probable cause of KPgME was the Chicxulub asteroid, which ejected enormous amounts of dust, ash, soot, and other aerosols at its impact. The outcome was prolonged sunlight screening, which resulted in months of global blackouts. The direct consequence was the extremely cold "impact winter," lasting for a decade. The exaggerated cooling destabilised all trophic levels in the biosphere, which resulted in the extinction of most species [20,21]. For parasites, host switching might have been an option to survive KPgME, as it is confirmed in the feather lice of birds [22]. In these hardly tolerable climatic conditions, freshwater microhabitats with higher thermal inertia could provide refugia for certain species. The detritus of these swamps and ponds supplied nutrients to the

survivors. In freshwater habitats, the biodiversity loss was barely 10%, contrary to the terrestrial 90% [20]. The recovery from this climatic catastrophe took about 30 years [20,21].

The seismic effect of the asteroid impact generated changes in the volcanic activity of the Deccan Traps, resulting in increased $CO_2$ outgassing [23,24]. Due to this extensive volcanism, the greenhouse effect augmented, and a drastic global warming began, which contributed to the formation of the early Cenozoic ecosystem [21,25]. The massive carbon release to the atmosphere finally led to the Paleocene-Eocene Thermal Maximum (PETM) around 56 MYA, characterised by global warming, reduced latitudinal temperature gradients, ice-free poles, and ocean acidification [26].

The emergence of the Fasciolidae family dates back to 90–100 MYA, after the fragmentation of Gondwanaland into Africa and South America [27]. Based on molecular genetic analysis, it is confirmed that the lineages of *Fasciolopsis* and *Fasciola* diverged cca. 88.1 MYA, long before the KPgME [28]. The recent intermediate hosts of the Fasciolidae family are gastropod species of the Basommatophora suborder, which evolved in the Carboniferous period. Based on the internal transcribed spacers of the ribosomal DNA (ITS-1 and ITS-2) and the mitochondrial 16 S ribosomal DNA, it is confirmed that Lymnaeidae snails, the most important intermediate hosts of the Fasciolidae family, originated during the Jurassic period [29], while the Lymnaeidae and Planorbidae families diverged 250 MYA [30], long before KPgME. Therefore, these snails are also survivors of KPgME.

Both Fasciolidae and their basommatophoran snail hosts evolved before the extreme winter caused by the Chicxulub asteroid impact. They also avoided extinction later, during the extreme hot climate of PETM. Presumably, their characteristic habitat, freshwater ponds and swamps, mitigated the temperature extremes and provided nutrients for the snails and thus their parasites. The harsh climate change caused the aridification of inland habitats; therefore, surface water cover decreased, resulting in habitat fragmentation [20].

Among these circumstances, isolated subpopulations, living in ponds, creeks, ditches, and shallow lakes almost worldwide, sustained the survivor species in small subpopulations [31]. Extreme habitat fragmentation jeopardises the sustainability of the subpopulations, thus the whole population itself, because if a population decreases below the effective population size, its maintenance is questionable. The ability of self-fertilisation reduces the effective population size, whereas hypothetically only one mature individual can multiply itself [13,32]. Several taxa of Basommatophora possess the ability of self-fertilisation, which assures the survival of the population after large-scale losses and contributes to the invasive ability of certain species [33,34].

The gene flow between the subpopulations of the habitat fragments is indispensable to the survival of the species. Lymnaeid snails are very small animals with limited moving ability; therefore, they cannot travel very far under their own power. However, these species are maintained during the rapidly changing KPgM; moreover, they spread worldwide from their fragmented refugia [29]. The historical spread of Lymnaeid snails is confirmed by the finding of a fossil *Galba* sp. from the Cretaceous period on an island 1500 km from the mainland. The migration route cannot be reconstructed; however, the authors presume that an ancient bird should have transported it on its body surface or within its intestinal tract through the Indian Ocean [35]. The same phenomenon is observed in the European distribution of *F. magna*, whereas ungulate wildlife can transport snails by mud stuck on hair [36]. This route can contribute to the translocation of a limited number of snails. Population-level spread of these small gastropods can occur in lotic habitats in the flow direction [37], which is generated by strong currents, e.g., stormwater runoff [38]. The phenotypic plasticity that helped these mollusc taxa survive and spread during the climatic extremities of the Cretaceous-Paleogene Boundary (KPgB) has been conspicuous recently. These snail taxa can be found in temperate zone floodplains [3,5,7,36], high-altitude mountain lakes in the Alps [39], in the Andean areas of South America [40], and in the tropics [41–43].

## 2.2. Distribution by the Proboscidea Order

Within the intensely changing environment between KPgB and PETM, the Fasciolid parasites should have met their ancient final hosts, the early proboscideans [1,27]. After the loss of the late Cretaceous megafauna, the ancient mammals began to diversify and fill the ecological niches left by the extinct non-avian dinosaurs [44]. The order of Proboscidea was one of the earliest known modern placental orders in the Late Paleocene, cca. 60 MYA in Africa [45]. The oldest fossils of ancient elephant relatives can be traced back to 45–60 million years [45–47]. Due to the drastically increasing temperature, tropical and subtropical forest environments became the dominant habitats globally, and the primitive proboscideans lived in swamp ecosystems and fed on freshwater vegetation in riverine or swampy settings [30,44] such as the Lymnaeid snails at that time [31].

The following 40 million years in the Cenozoic Era witnessed the evolutionary history of proboscideans. During this time period, the order diversified and spread almost all over the world through three major radiations [28,46]. The first occurred during the Palaeocene Eocene Boundary (56 MYA), with the diversification and intra-African spread of primitive proboscideans [45,46]. The second radiation took place during the Oligocene and early Miocene periods, when the supervening connection of Africa to Eurasia could facilitate the faunal interchange between the two continents. This eventuated 25–20 MYA and was termed the Proboscidean Datum Event (PDE), which resulted in rapid global dispersal of the order, by which it reached the Americas [46,48]. After the PDE, the proboscideans became one of the most diverse taxa of the paleofauna [48]. The last radiation took place during the late Miocene/early Pliocene and resulted in the diversification of Elephantidae, the ancestors of the extant two genera, *Loxodonta* and *Elephas* [46].

The phylogenetic analyses of the Fasiolidae family support the hypothesis that proboscidean dispersal spreads these parasites globally. The basal extant member of the taxon, *Protofaciola robusta*, parasitises the small intestine of the African bush elephant (*Loxodonta africana*) and uses Planorbid snails as intermediate hosts. The next most basal member, *Fasciolopsis buski*, is also an intestinal parasite with a planorbid intermediate host [1]. The whole genome analysis of Fasciolidae revealed that the clade comprising *Fasciola* and *Fascioloides* genera split from the ancient lineage between 65 MYA and 55.9 MYA during the rapid diversification of the Proboscidea order. This split also meant a switch between both habitats (from the intestine to the liver) and intermediate hosts (from Planorbidae to Lymnaeidae) [1,27,28].

The closest relative of *F. magna* is *Fascioloides jacksoni* [49], which parasitises the liver of the Indian elephant (*Elephas maximus*). This fact suggests that these sibling species were spread by their primary definitive hosts, which were proboscideans [50,51]. A comparative study of karyotypes and chromosomal locations of rDNA genes in *F. magna* and *F. hepatica* suggested that *Fasciola* is the younger genus within the liver pathogen clade of Fasciolidae [52]. Based on molecular genetic investigations, fossil records, and recent epidemiological and pathological findings, it is very probable that the *Fasciola* genus evolved as an adaptation to the emerging taxon of ruminants [27]. Though a recent phylogenetic investigation hypothesised that the oldest member of the *Fasciola* genus is *F. nyanzae*, the first hosts of this genus were hippopotamuses, and subsequent fasciola species jumped to ruminants by an intermediary host, which could be the wild boar (*Sus scrofa*) [53].

## 2.3. The Fall of the Proboscideans and the Dawn of the Ruminants

After the PETM, about 34 MYA, the temperature fell and the aridification process started. During the Miocene (23.03–5.333 MYA), the climate began to cool drastically, which induced changes in the ecosystem. The cooler and, in mid-latitudes, dry and seasonal climate supported grasslands to spread. The early fossil record of grasses dates back to the Late Cretaceous; however, until the Miocene, grasses did not form extensive grassland ecosystems [44]. The previously woodland-dominated vegetation turned to more open, savanna-type environments [54,55].

This period coincided with the Messinian Event 6.0-5.3 MYA, when the Mediterranean dried up and the world's oceans became less saline [56]. Just after that, the Miocene-Pliocene Boundary (MPB, 5.3 MYA) marks a transition from a late Miocene cooling trend to the early to middle Pliocene warm period, when the Northern Hemisphere was largely ice-free and atmospheric $CO_2$ concentrations were comparable to present-day levels [57]. This interim warming during the global cooling trend turned to the subsequent glacials and led to the unstable Pleistocene with extended arid glaciations, sea level changes, fluctuating atmospheric carbon dioxide, and shifting vegetation [56]. During this volatile geological time of MPB, the two sister species of the genus *Fasciola*, *F. hepatica* and *F. gigantica*, diverged [28].

The alteration in vegetation brought about a dramatic change in the nutrient supply of the megafauna. Those taxa could reach evolutionary success, which adapted to the use of grasses as nutrient sources (grazers, GR); while those herbivores, which depended on woodland plants (browsers, BR), began to decline [48]. The diversification of GR and the decline of BR were initiated during the Miocene and accelerated through the MPB. This process was more rapid in North America than in the rest of the world [55].

Among these conditions, ruminants began to diversify extensively through adaptation to more heterogeneous habitats and seasonal climates, and they colonised a large range of biomes [54]. In the meantime, specialist biodiversity declined due to reduced primary production and the loss of forest habitats. Proboscideans are mostly browser herbivores, which means that they depend on woody and non-woody dicotyledonous plants, leaves, and barks of trees as food sources [44,56]. Owing to the enhanced competition with other ungulate clades that evolved specialist grazing ecomorphs, the extinction of the proboscidean species accelerated around 8 MYA, after the onset of grassland-dominated habitats [48,55].

The global temperature and the consequential vegetation shift cannot explain the replacement of proboscideans with grazing-adapted ungulates [44,54]. A paleontological study confirmed that adaptation to grazing lifestyles could be detected in late gomphotheres, a proboscidean taxon that went extinct in the late Pleistocene [58]. However, another study highlighted that proboscidean fossils from the middle Miocene to the late Pleistocene showed enamel hypoplasia, which can be regarded as a sign of chronic stress, though the reason cannot be specified [59].

Multiple extinction events occurred during the Late Pleistocene, mainly over the last 100,000 years, with a peak between 12,000 and 10,000 years ago [44,55]. The most severely affected fauna was large-bodied herbivores above 1000 kg body weight. Especially in the Americas, where 100% of this sized species became extinct [55]. This era of natural history is characterised by the human presence. A consensus is emerging that climate change and early human activities went together in contributing to the biodiversity loss of megaherbivore fauna [44]. The top-down effects of human predation, landscape modification, and especially the controlled use of fire affected the proboscidean populations of the palearctic regions [55,58].

In contrast with proboscideans, the ruminants proved to be the real winners of the climatic fluctuations of the last 45 million years. After the Eocene-Oligocene cooling and aridification event 34 MYA, and especially during the Miocene Climatic Optimum cca. 18–15 MYA, the ruminants diversified and successfully adapted to the heterogeneous habitats and climate of the Earth [54].

*2.4. Switch to the Cervid Host*

Within the Ruminantia suborder, the Bovidae and Cervidae families evolved in Asia during the Miocene and started colonising a large range of ecosystems [54,60]. By analysing two mitochondrial protein-coding genes and two nuclear introns for 25 species of deer, an investigation estimates that the Cervidae family originated in the Late Miocene, 7.7–9.6 MYA, and the common ancestor of the American Odocoileini is dated between 4.2 and 5.7 MYA. The fossil records of American cervid species also supported the molecular findings that

Odocoileini entered the American continent during the MPB, while other cervid taxa reside in America probably as a result of a recent dispersal event, cca. a million years later, during the Early Pleistocene [60] or at a much later time, during the presence of the last landbridge between Eurasia and America between 10,000 and 70,000 years ago [61].

The host switch of *F. magna* from proboscideans to cervids presumably eventuated in North America [1]. In Eurasia, a large number of potential host species were available, and the proboscidean loss was not as severe as in the Americas [27,54,55]. It is very probable that a drastic population collapse of the American proboscidean host species forced the host shift to the sympatric cervid species [1], which could be the ancestor of the recent white-tailed deer (*Odocoileus virginianus*).

On the North American continent, white-tailed deer were the most abundant host species for thousands of years. It is estimated that before the arrival of the European settlers, between 23 and 40 million white-tailed deer inhabited North America. Due to its extensive hunting, cca. 500,000 animals remained by the late 1800s. During the 20th century, 41 states made a law for the protection of the species; therefore, the currently estimated population is between 14 and 20 million individuals in the United States [3].

On the North American continent, *F. magna* has five focuses: (1) the Great Lakes region; (2) the Gulf coast, lower Mississippi, and southern Atlantic seaboard; (3) the northern Pacific coast; (4) the Rocky Mountain trench; and (5) northern Quebec and Labrador [62]. Though potential habitats and appropriate intermediate and final host species exist even between the main focuses, considerable expansion cannot be detected [7]. Phylogenetic analysis of *F. magna* strains originating from different American focuses confirmed that most populations share haplotypes with low genetic differences; however, the eastern and western populations show genetic distance [63,64]. The patchy distribution and the contradictory genetic overlap of the geographically separated subpopulations are suggested to be the consequence of the final host's population loss, which forced the partial switch to alternative hosts belonging also to the Cervidae family. This phenomenon is frequently detected in host-parasite systems during host extinction when the parasite prefers host-switching to species that are phylogenetically close to its original host [65].

## 3. Recent European Distribution

The European history of *F. magna* proves substantially eventful. After the first introduction, which remained a local endemic in La Mandria, later immigration events resembled a territorial conquest. Importation of infected white-tailed deer and wapitis from endemic areas is supposed to be the main route of disease introduction to Europe [3,5–7]. The parasite seemed to switch to Eurasian cervid species without impediment, presumably due to the phylogenetic proximity of the previous and new hosts [3,65]. The exact beginning of the process can hardly be determined, notwithstanding two further focuses formed in Central Eastern Europe during the 20th century [3,5,7]. One of those emerged in Czechia and expanded to Poland, or vice versa, whereas Poland also imported wapitis from the United States during the 19th century [66]. The occurrence of the parasite is estimated to have happened during the first third of the 20th century in Czechia [67,68].

The other focus formed in the floodplain forests of the Danube basin. The first cases were detected in Slovakia by the Hungarian border [69], then a year later on the opposite side of the border river Danube in Hungary [70,71]. This ecosystem provides ideal conditions for continuous spread through the continent by the river's flow direction [3,5,7,72,73].

In Croatia, *F. magna* occurred around the turn of the millennium in the Kopacki Rit Nature Park at the confluence of the Danube and the Drava [74–76]. Some years later, during the 2007–2008 hunting season, its presence was reported from Serbia, near Belgrade at the Danube-Sava confluence [77,78]. The species was recorded in the early decades of the 20th century in Germany for the first time [79]. In Bavaria, a suspicious case was detected in 2009 near the Czech border, and the endemic was confirmed later, in 2011 [80]. In its Austrian distribution area, the first confirmation of the parasite happened in an enclosure in 1981 [81]. However, in free-range territories, *F. magna* established itself almost 20 years later,

near Vienna [15]. The relevance of tributary streams of the Danube as potential expansion routes has been confirmed several times. In Slovakia, the Vah/Vág and the Hron/Garam open gates to spread opposite the flow direction [73]. By the Croatian-Hungarian border river Drava, the same phenomenon can be observed [75,76,82,83].

Molecular investigation of the mitochondrial genome of *F. magna* confirmed that the North American population possesses a higher level of genetic diversity, while the European populations belong to two phylogenetic lineages [5,64,84,85]. The investigations revealed that the population of the Italian focus originated from the northern Pacific coast, while the two other European focuses acquired infection from South Carolina: the Gulf coast, lower Mississippi, and southern Atlantic seaboard focus (Figure 4) [5,7].

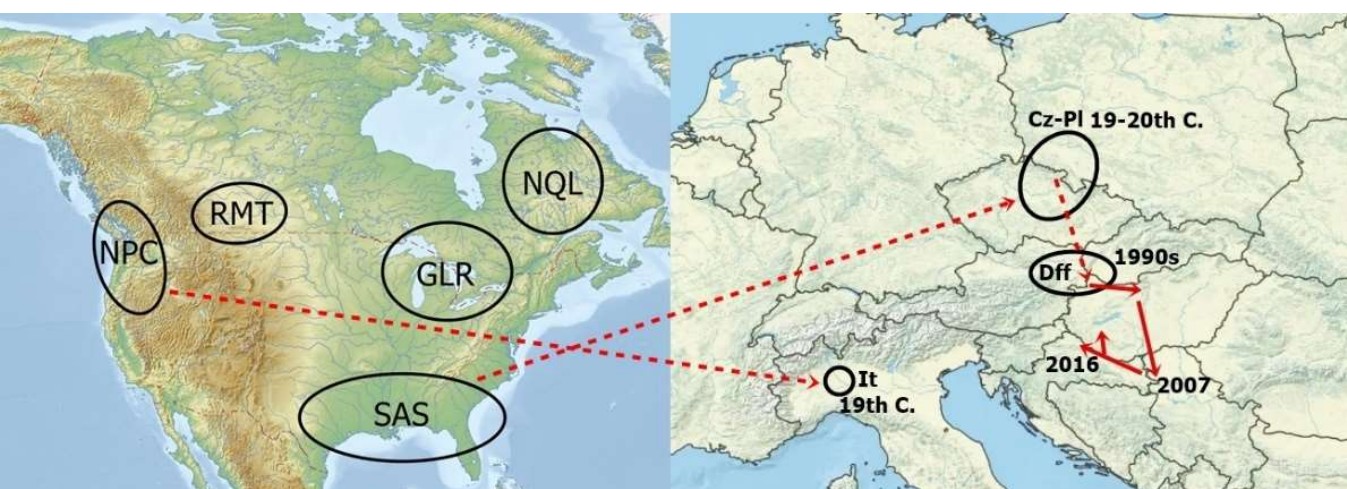

**Figure 4.** Natural (North American, on the left) and acquired (Europe, on the right) areas of *Fascioloides magna* with the time of the most probable occurrence of the parasite. Continuous arrows = natural distribution; Dashed arrow = human-mediated distribution cannot be excluded. NPC: Northern Pacific Coast, RMT: Rocky Mountain Trench, NQL: Northern Quebec & Labrador, GLR: Great Lakes Region, SAS: Southern Atlantic Seaboard.

## 4. Final Hosts of *F. magna*

In the development of *F. magna*, several mammal species can serve as final hosts. Based on the tissue lesions of the liver, the progress of the disease, and the role in maintenance of the parasite, the final hosts are sorted into three groups: definitive, dead-end, and aberrant hosts [3,5,7]. In its natural distribution area, North America, the ancient host of the parasite was white-tailed deer (*Odocoileus virginianus*). However, wapiti (*Cervus elaphus canadensis*) and caribou (*Rangifer tarandus*) can also maintain the populations of the fluke. Other cervid species, such as black-tailed deer (*O. hemionus columbianus*) and mule deer (*O. hemionus hemionus*), have lower relevance [85]. Domestic animals cannot contribute to the maintenance of a *F. magna* endemic, whereas in livestock, the parasite finds itself at a dead end. In horses [86], pigs [87], and cattle, the flukes can mature and produce eggs, which cannot reach the environment because of the histopathological structure of the liver lesions. In domestic small ruminants, such as sheep and goats, the infection generally kills the animals before fluke maturation; therefore, these animals are classified as aberrant hosts [3,7,85].

Within the acquired areas of Europe, red deer (*C. elaphus*) and fallow deer (*Dama dama*) are confirmed to be definitive hosts. Smaller-sized ruminants, such as roe deer (*Capreolus capreolus*), mouflon (*Ovis aries musimon*) [3,7], chamois (*Rupicapra rupicapra*) [88,89], and domestic sheep and goat, proved aberrant hosts based on the primary experiences of the European endemics [3,7].

After entering the small intestines of the final host, the cercariae penetrate the intestinal wall and begin their migration through the abdominal cavity towards the liver. At a very early stage of infection, milk spots and the signs of capsule perforation can be seen

during the necropsy. Within the liver parenchyma, the developing flukes are looking for potential mating partners, causing characteristic lesions of larval migration. These lesions are brownish stripes (Figure 5), which contain haematin, a product of red blood cell disintegration as a consequence of tissue and bloodvein damage caused by the juvenile parasites migrating through the parenchyma [7,9]. As the flukes grow, the diameter of the migration channel also increases [72].

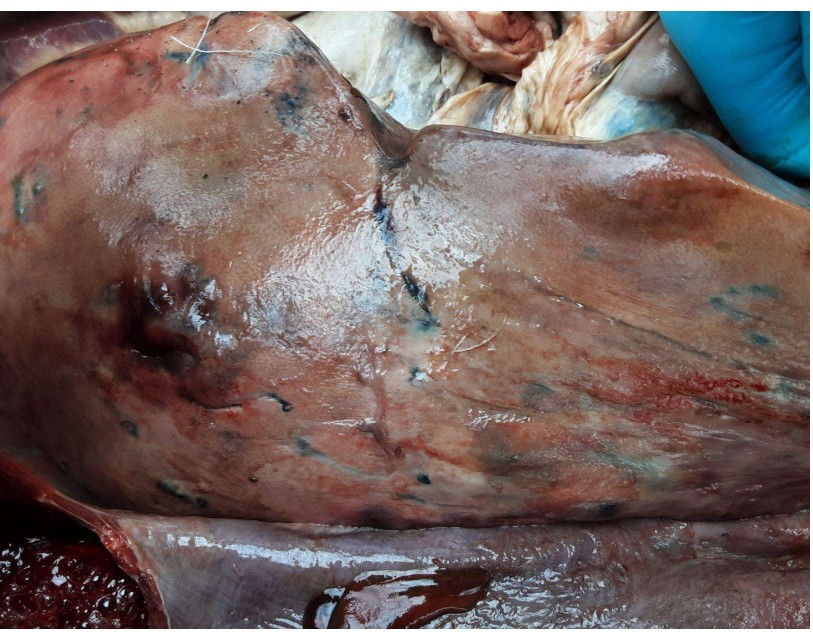

**Figure 5.** Brownish stripes on the surface of an enlarged liver are a sign of the juvenile fluke's migration.

The definitive hosts behave in the parasite's life cycle very similarly to the white-tailed deer, the native American final host of the large American liver fluke. In this type of host, the liver capsule is generally covered by a thick fibrin coat with a jagged or filamentous surface, which sticks the organ to the diaphragma. Due to the juvenile fluke's migration, different diameters of migration tunnels can be seen both on the liver surface and in the deeper tissues [72]. When the parasite finds a partner or gives up further searching, it terminates its migration. As a result of the parasite migration, excessive hepatocyte damage causes fibroblast proliferation, thus progressive fibrogenesis, which leads to the formation of thick-walled pseudocysts (Figure 6) around the parasites [8]. The wall of the pseudocyst contains connective tissue and smooth muscle elements, while inside the cyst, brownish-greenish detritus and 1–3 parasites can be found [3,7,8]. Histopathological investigation of the remnant parenchyma demonstrates general chronic inflammation, infiltration by macrophages and eosinophils, degeneration of hepatocytes, and wall thickening in both blood veins and bile ducts [8,10]. In some carcasses, secondary bacterial infection of the necrotic tissue occurs, which is indicated by the yellowish colour of the lesion or haematin mixed with blood [72]. Some of the bile ducts are incorporated into the pseudocysts, thus the mature flukes can shed eggs through this route [7]. In the advanced stages of infection, extensive fibrosis of the whole parenchyma occurs, which causes the deformation of the organ [72]. In definitive hosts, the prepatent period is at least 30 weeks, and the lifespan of the flukes can reach five years [3]. In Europe, red deer (*Cervus elaphus*) and fallow deer (*Dama dama*) are confirmed to be definitive hosts [3,7], while the assessment of roe deer (*Capreolus capreolus*) has been varying recently [82,83].

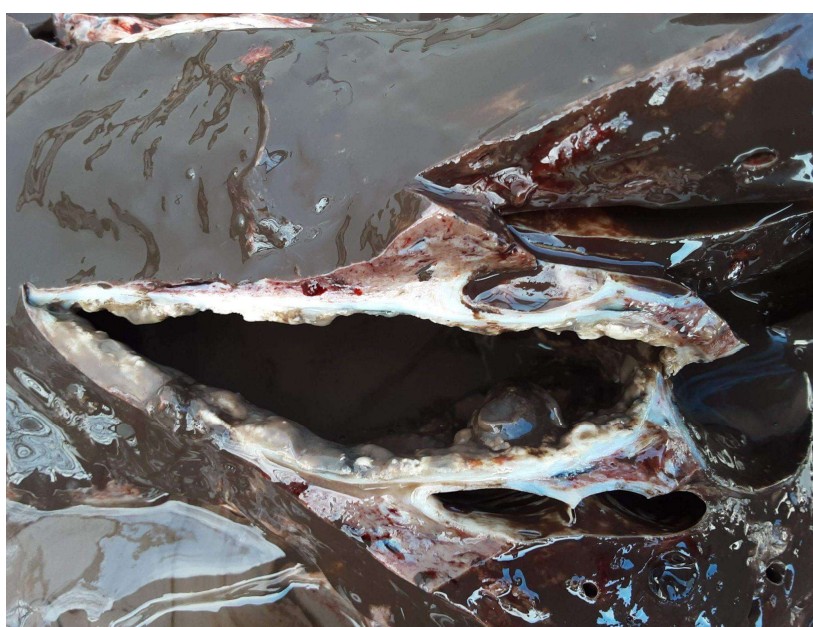

**Figure 6.** An extremely large, thick-walled pseudocyst in a red deer's (*Cervus elaphus*) liver parenchyma contains a huge amount of greyish-brownish detritus.

In dead-end hosts, the parasite may reach sexual maturity similarly to definitive hosts. However, the host's body reacts differently than of the definitive hosts. The wall of the pseudocyst is very thick, and no exit is created by bile duct incorporation; therefore, in dead-end hosts, the parasites are ambushed, and eggs cannot fulfil their destiny. These hosts generally do not show severe clinical signs of fascioloidosis [5,90]. Dead-end hosts can shed eggs when a mechanical trauma injures the pseudocyst, and the eggs can reach the bile ducts, thus the intestinal tract and the environment [3]. In Europe, wild boar (*Sus scrofa*) is the most important dead-end host in natural habitats. It is suggested that hunting offals containing pseudocysts with eggs can pose a risk for parasite spread, especially if these remnants are used by carnivores [5]. In the Croatian territory of *F. magna* distribution, Konjevic et al. experienced that wild boar can develop thinner-walled pseudocysts, similarly to definitive hosts [89]. Among farm animals, cattle, pigs, and horses can play the role of dead-end hosts [3,7]. In heavily infected American areas, decreased performance and a high rate of abattoir liver condemnation are observed in pasture-reared cattle [5]. In Europe, a similar phenomenon has not been confirmed yet [3]. It should be noted that the determination of the infection prevalence in bovids is hampered by visual-only meat inspection at abattoirs. The lesions of the liver, which are supposed to be abscesses, are not incised to avoid potential contamination of the meat. Therefore, *F. magna*-origin lesions in bovid carcasses might be under-detected by routine meat inspection [6].

Neither aberrant hosts contribute to the maintenance of *F. magna*. In their body, the parasite cannot reach sexual maturity. After the infection, a very excessive wandering of juveniles occurs, which causes severe tissue damage in both body cavities. The haematin containing greyish-brownish stripes of migration tracks can be seen in the liver, under the serosa of the visceral organs and the abdominal cavity, and also in the lungs and under the pleura. The tissue degeneration causes multiple organ failure, which leads to the death of the animal before the flukes mature, mostly 4-6 months after infection. Pseudocyst formation in aberrant hosts cannot be detected [3,7]. Among the aberrant host, mostly smaller-sized ruminant species, such as mouflon (*Ovis aries musimon*), sheep, goat [7], and chamois (*Rupicapra rupicapra*) [88,89], can be found.

### 4.1. The Special Role of Roe Deer

The roe deer (*Capreolus capreolus*) was considered an aberrant host, whereas a severe, frequently fatal course of infection could be observed in this host, especially after the emergence of *F. magna* in a new area. In these cases, necropsy investigation of the carcasses did not detect pseudocyst formation [3,7,15] or egg production [72] in roe deer. Currently, increasing amounts of evidence confirm that in roe deer, pseudocyst development [91], egg production [82,83] and egg shedding [92] are common phenomena in endemic areas of Europe.

In roe deer, the macroscopic lesions of fascioloidosis differ from those in other definitive hosts. The fibrin coat on even chronically infected livers is thinner than in red and fallow deer. On the other hand, migration tunnels of juvenile flukes occur as linear depressions on the surface of the organ. The number of adult worms in a carcass and per cyst is lower in roe deer than in other cervids [72]. This phenomenon can also be detected in white-tailed deer in American distribution areas [74].

Among potential definitive hosts in Europe, roe deer is the only one with a solitary lifestyle. These animals live, feed, and drink in a limited territory; they encounter their conspecifics during the mating season, so the acquisition of infection is incidental [82,91]. In different areas, a very diverse prevalence can be detected. In Czechia, 70–80%, in Slovakia, 60% [7], in Hungary, 36.7% [83], and in Poland, 13.6% [72] prevalence was confirmed in roe deer. This also supports the hypothesis that roe deer data cannot reflect the real epidemiological situation in a region [91]. Different roe deer territories represent different habitats with varying epidemiological risks of *F. magna* infection [82].

However, roe deer, also belonging to the Capreolinae subfamily, is a closer relative to white-tailed deer than any other Old World Cervidae [60]. Considering that parasites' host switch can occur most commonly between species in phylogenetic proximity [65], roe deer was expected to serve as an efficient host for *F. magna* to fulfil its life cycle. The higher mortality rate is presumed to be the consequence of the small body size of this species, which barely accounts for the damage caused by large-sized parasites. This hypothesis is supported by the fact that fallow deer, the next smallest Eurasian cervid species, also show a significantly high loss during *F. magna* infection, though it is considered a definitive host based on its epidemiologic role and the pathological course in the liver parenchyma [3].

However, in areas where the infection has been established for years, a host-parasite co-evolutionary process is observable. In the Croatian distribution area, after 20 years of presence, the investigated 34 roe deer showed chronic infection and the signs of reinfection in 20%. This means that a considerable proportion of the infected population does not die from fascioloidosis [82]. A very similar phenomenon is observed in a Hungarian area where the infection was detected first a few years before. In this investigation, the researchers found exclusively chronically infected roe deer in almost 40% of the examined 60 carcasses. Moreover, a considerable population loss in this study area was not detected as a result of *F. magna* emergence [83]. Both investigated endemics are part of the Danube basin distribution of *F. magna*, which has been occupied for more than two decades [7,68].

The very beginning of the European *F. magna* invasion was characterised by a strong impact on naive cervid populations. Reduced reproduction performance and trophy quality and increased natural loss are observable after the first emergence of the parasite [74,93]. After two decades of attendance in Central and Eastern Europe, *F. magna* appeared to lose its virulence. In an investigation in Bavaria 2019–2020, Sommer et al. determined a 5.9% prevalence in red deer (N = 640), of which 76.3% was mild infection. However, in wild boars, fallow deer, and roe deer, no infected individuals were detected by investigating 60 carcasses total. The authors concluded that the very low prevalence of the parasite in the examined red deer population is attributable to the high proportion of mild, thus barely noticeable cases. Based on their findings, the researchers suggested that *F. magna* was underdetected in its settled endemics [10].

At the European occurrence of *F. magna*, roe deer was evaluated as an aberrant host, whereas this small-sized cervid suffered heavily from the parasite. Fatal cases were com-

mon, and a serious population-dynamic impact was detected [3,5,7]. Recent studies on roe deer confirmed that the parasite and its host got into balance [82,83]. Since the first detection of the parasite in the Danube basin, only some generations of roe deer were born, and due to the territorial lifestyle of roe deer, the encounter between the populations of the host and the parasite was not as extensive as in the case of red deer. [82,91]. The adaptation of roe deer populations to the new parasite is less probable than vice versa. In a coevolutionary process, both the host and the parasite affect each other's population size, which alters the polymorphism in both populations. These cycles of coevolution are more detectable in parasites than in their hosts [94]. Those roe deer populations, which were surveyed in the 2020s, showed no significant loss, thus gene frequency alteration. It is suggested that the virulent strains of *F. magna* became the victims of the arms race between the host and the parasite, whereas they were lost together with the hosts they killed [83]. In its invasive range, a parasite must surrender its virulence in favour of population maintenance. This phenomenon is especially characteristic of generalist parasites, which are highly adaptive to heterogeneous environments [95].

### 4.2. Efforts to Control F. magna in Cervid Hosts

Experiencing the terrifying appearance of the liver lesions of infected animals, the hunting communities conceived the idea of medical control for the large American liver fluke. In the case of *Fasciola hepatica*, one of the closest relatives of *F. magna*, triclabendazole (TCBZ), proved to be the most efficient drug to control the parasite. This is the only drug that can kill both juvenile and adult worms. The generally applied dose of TCBZ in domestic ruminants is 10 mg/kg bodyweight [96].

For treatment of *F. magna* infection, the efficacy of TCBZ can reach even 100% [76]. In enclosures, 95.5% of the red deer showed recovery from facioloidosis after individual intraruminal application of the medication. However, in the same study, the authors reported 20–80% infection prevalence 30 days after the 60 mg/kg bodyweight TCBZ bait-treatment of free-ranging red deer [75]. In Bavaria [80] and Serbia [77], successful TCBZ-based control programmes were carried out in fenced areas. However, the authors noted that habitat interventions, such as drying or fencing of wetland and strict mowing of shores, were also relevant elements of integral parasite control [75,77]. For enclosures, besides the TCBZ treatment of stock replacement, a 30-day quarantine period can provide appropriate safety against *F. magna* importation [70,75].

In free-range systems, the efficacy of medical control is influenced by several environmental factors [76]. At the first occurrence of the parasite in the Danube flood forests, a TCBZ treatment campaign was carried out with the international cooperation of Austria, Slovakia, and Hungary. Though a temporary improvement could be detected, the endemic was maintained, and the spread could not be impeded [5]. In Austria, the control programme was continued with 10–15 mg/kg feed TCBZ application twice a year. After 2010, the parasite was claimed to be endemic in the country along the Danube [97].

Recently, a consensus suggests that medical treatment of free-ranging populations can improve the epidemiological situation; however, it proves unsuitable for eradication [3,76,91]. The removal of intermediate hosts arose at the very beginning of research on *F. magna*. Copper sulphate proved efficient against the snail hosts of the parasite. However, this method was already dismissed at that time. Anyway, its relevant ecological impact, it was overly expensive to apply it to extended areas [6]. For the protection of domestic livestock, both the older [6] and the current literature [75] suggest isolation between wildlife and pastures for domestic Bovidae.

## 5. Intermediate Hosts of *F. magna*

Current intermediate hosts of *F. magna* are referred to the gastropod suborder Basommatophora, which comprises more than 300 species in five families. The most abundant families of this taxon are Lymnaeidae, Planorbidae, and Physidae, involving 90% of all species within Basommatophora [29]. The Planorbidae family is the largest, comprising

250 species. Lymnaeidae and Physidae families have cca. 100 and 80 species, respectively [98]. These families possess several intermediate host species for other digenean Trematodes with human health and veterinary relevance [29].

On the American continent, six snail species proved to serve as an intermediate host of the large American liver fluke, i.e., *Lymnaea modicella*, *L. caperata*, *Galba* (syn. *Lymnaea*) *bulimoides*, *Fossaria* (syn. *Lymnaea*) *parva*, *Stagnicola* (syn. *Lymnaea*) *palustris nuttalliana*, and *Pseudosuccinea columella* [3,5,7,11,12]. However, in Europe, only two species—*Galba truncatula* and *Radix labiata/peregra*—are confirmed to be able to acquire infection and produce viable cercariae in natural circumstances [5,11,73].

The identification of susceptible intermediate host species presents significant difficulties. On the one hand, infection prevalence in snail populations, even within confirmed endemics, is very low—less than 1% [15,97,99]. On the other hand, the appropriate morphological identification of snails is almost impossible, and their phylogenetic classification changes time and again [29,100,101].

During the history of malacological research, 1143 species of the Lymnaeidae family were described; however, fewer than 100 species remained after the synonymization of identical ones. Most genus names, except for *Radix*, are based on phenotypic resemblance instead of sound evolutionary and phylogenetic considerations [29]. Within the *Radix* genus, *R. peregra* and *R. labiata* proved to be conspecifics, based on karyotype analysis [102] and molecular genetic investigation of internal transcribed spacers in the ITS-2 sequence [90,100,101]. Among the proven American intermediate host snails, *Lymnaea modicella* and *Fossaria* (syn. *Lymnaea*) *parva* are suspected to correspond to *G. truncatula* species [29]. Recently, molecular methods have been developed for proper species identification of potential intermediate host snails [40,103–105].

Based on the two nuclear internal transcribed spacers of the ribosomal DNA (ITS-1 and ITS-2) and the mitochondrial 16 S ribosomal DNA of 38 Lymnaeid species, Correa et al. demonstrated that the Lymnaeidae family comprises three separate phylogenetic clades, of which C1 originated on the American continent and incorporates the genus *Galba*. The two other clades are both Old World-originating; C2 is Eurasian, while C3 is Australian-Indopacific-originating, and the latter incorporates the *Radix* genus, with the most important intermediate hosts of *F. gigantica*. This investigation proved that the recently Eurasian-distributed *G. truncatula* diverged from its New World sister species 145–200 MYA [29].

*5.1. Susceptibility of the Intermediate Hosts*

Among the known intermediate host species, susceptibility and cercaria-producing efficiency are not identical. In Europe, *G. truncatula* provides a more suitable environment for the development of the flukes than *R. labiata*; however, the latter is more abundant and can tolerate acidic environments, therefore its distribution is wider [3,5,7,11,90]. In the case of *F. hepatica*, the ability of *G. truncatula* and *R. labiata* to act as intermediate hosts and produce viable cercariae proved similar [106], though a previous study confirmed the opposite [107]. In two further potential lymnaeid hosts, *Radix ovata* and *Stagnicola palustris*, the larval development breaks off and the snail perishes [3]. Though in *S. palustris*, *F. hepatica* can complete its development [108].

Within a certain species, individual differences can be observed, as bigger, well-developed snails can produce more cercaria [3,12,15], and older snails are more resistant to the infection [3,7], while experimental infection of snails with miracidia originating from a different region resulted in slower cercaria production [7].

The success of infections is also affected by environmental factors. In Austrian habitats, stony soil with poor vegetation is a place where the prevalence of infection is the lowest; however, the snail density is also very low. In the same investigation, the researchers found that the infection reaches its peak in June and July when the water temperature is between 20.6 and 20.8 °C, though the optimum for the miracidia (Figure 7) to infect snails is 22–25 °C [97]. At very high summer temperatures, the snails aestivate, reducing their

metabolism to avoid exsiccation and heat shock. Therefore, in hot weather, the density of detectable snails reduces. As a consequence, spring-born snails begin reproduction during the fall, and their offspring overwinters and reproduces by the next May–June [3].

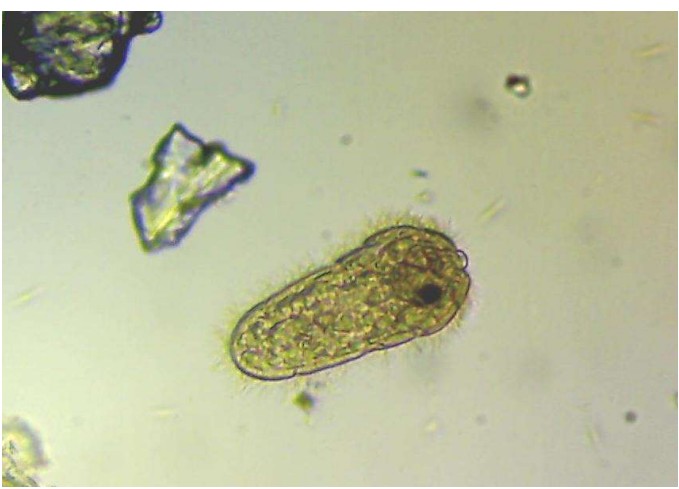

**Figure 7.** Free-swimming miracidium after hatching: besides cercaria, this developmental stage of *Fascioloides magna* can be found in the environment when ambient temperature is between 22 and 25 °C.

*5.2. The Course of Infection in Intermediate Hosts*

In the body of the intermediate host, asexual multiplication takes place. The miracidia, which penetrate the foot or the mantel region of the snail, after reaching the pulmonary sac, transform into sporocysts and produce mother rediae. Mother rediae migrate to the hepatopancreas and rear daughter rediae, which leave the mother rediae, and the process is repeated again. The cercariae emerge in the daughter rediae and leave the snail's body [2,3]. This process lasts for 6–7 weeks [15]. As a result of this asexual multiplication, from one miracidium, 164–210 [12] or even 1000 [62] cercariae could be produced. The cercariae occur in the environment between May and October, allowing the infection of final hosts during this period of the year [109].

Being infected with trematode larvae, the snails can show signs of functional disorders of the kidney, the gonads, and the digestive glands [110]. Mostly, they decrease or even stop reproduction. In pre-adult snails, the infection can completely destroy the genitals [3]. In adult snails, or in more resilient species, e.g., *P. columella*, the reproductive disorder can be temporal [11]. In pre-adult snails, the development pattern changes, as both parasitic gigantism and decreased shell growth can be observed [7,12]. As a consequence of their extensive immune defence, the snails depleted their reserves, which resulted in a population decrease some weeks after the infection [11]. During the immunological activities, the snails' hemocytes and their pattern recognition receptors, such as C-type lectins and toll-like receptors (TLR), play a central role. These cells can phagocyte trematode larvae. By recognising pathogen-associated molecular patterns (PAMPs), C-type lectins mediate the immunological process, which enhances phagocytosis [41] and modulates the humoral response [13]. In the case of trematodes, the PAMPs are carbohydrates, e.g., mannose, which take place on the surface of the sporocyst [111]. The hemocytes' TLRs, the primary modulators of the innate immune response, initiate the proinflammatory signalling pathways, which result in an acute phase response and redox killing, contributing to an effective cellular cytotoxic immune response [41].

The efficiency of this process shows differences both intra- and interspecifically. *Pseudosuccinea columella*, which was confirmed to be an intermediate host of *F. magna* in America, tolerates the infection with the parasite [11,112], while the members of the *Galba* genus can survive the experimental infection in 30–60%, depending mainly on the snails' size [11,12]. The intraspecific resistance against trematodes was studied in *P. columella*. In

some regions of Cuba, a *F. hepatica*-resistant strain was detected. These snails have some morphological alteration, a different pattern of the mantle; however, only a 0.17% genetic difference can be identified. The resistant populations are highly adaptive to the environment as well. They can live in soft water with an acidic pH where the mollusc diversity of the ecosystem is low, which mitigates the competition with other snail populations. Their resistance to *F. hepatica* required a fitness cost, whereas they possess lower fecundity rates, delayed egg hatching, and thus a diminished net reproduction rate. The rearrangement of the resources from perivitelline fluid production to immune functions is supposed to be in the background. The *F. hepatica*-resistant *P. columella* strain did not prove to be resistant to other trematode larvae [41].

Rondelaud et al. experimentally infected *G. truncatula* snails, which originated from four French habitats. They investigated the survival rate and cercaria production after the *F. magna* infection. They confirmed that significantly different survival rates can be detected both between populations and between certain individuals. The authors ascertained that small-sized individuals can produce fewer cercariae after infection with a single miracidium than their larger-sized counterparts [12].

On the American continent, the intermediate hosts of *F. magna* seem more resilient to the infection, presumably due to the long-lasting co-evolution with the parasite, contrary to their European counterparts, which were given only decades for adaptation [3]. On the other hand, observations with *F. hepatica* prove that parasite infection affects the local snail populations most severely, while immigrants' survival rate is higher [113].

### 5.3. Self-Fertilisation: A Trade-off between Population Persistence and Population Health

The worldwide distribution of basommatophoran snails is a result of successful recovery after population losses. The most important European intermediate host of *F. magna*, *G. truncatula*, has the ability to reconstitute the population with less than 10 survivors [114]. This extremely low effective population size is due to the high, about 90%, selfing rate of the species [37,115]. By self-fertilisation, the population surrenders the benefits of the outcrossing mating strategy, risking genetic load. Due to extensive inbreeding, invasive populations of these snail species have lower genetic diversity than their native populations in their original distribution territories [116,117].

The recurrent bottleneck effects decrease the genetic variability of the concerned populations. This is supposed to cause vulnerability to environmental factors; however, *G. truncatula* reached a wide-spread distribution with its heterozygote deficiency [32,37]. A study investigating *Galba truncatula* confirmed that a very low, 2% rate of outcrossing did not result in demographic consequences of genetic load in this species [118]. The same phenomenon can be observed in other invasive species, such as water hyacinth (*Eichhornia* syn. *Pontederia crassipes*) and *P. columella*, of which the invasive populations possess significantly depleted genetic diversity [32,119].

On the other hand, increased susceptibility to pathogens can be detected as a result of inbreeding. Highly inbred snail populations show an exaggerated predisposition to trematode infection; therefore, a higher infection prevalence can be observed in them, which results in a higher infection risk in the final hosts [13,32]. *Galba truncatula* and *P. columella*, a lymnaeid snail with similar selfing ability, prove more effective hosts for parasites than the locally distributed non-invasive snail species [32,119]. A comparable phenomenon can be noticed in *Biomphalaria pfeifferi*, in Senegal. A dam construction, which altered the water salinity of the Senegal River, selected a highly inbred population of *B. pfeifferi*. Before the dam was constructed, only 1% of the snails belonged to this species. After the completion of the barrages, 70% of the snails were *B. pfeifferi*. Whereas most snails within the affected river section showed high susceptibility to parasite infection, *Schistosoma mansoni* trematode began a population expansion in the region, causing human health risks [120].

### 5.4. Environmental Demands of Snail Hosts

The geographical distribution of the potential snail host can determine the occurrence of trematode parasites [13], so knowledge of environmental demands is indispensable to epidemiological research. The most relevant requirements of lymnaeid snails have not changed since KPgB [66 MYA] [20,31]. These snails prefer lentic or slowly flowing freshwater habitats with easily warming water along the shallow shorelines [121,122]. However, the generalist members of the taxon, e.g., *G. truncatula* and *R. labiata*, can be found in Alpine regions as well [39,101,123]. They prefer increased availability of nutrients, which is assured by progressive eutrophication (Figure 8) as a consequence of human disturbance [39,121]. They hardly tolerate acidic pH [5] because this condition, especially below pH 6, disturbs shell development. However, with a high calcium supply, the disadvantageous effect of acidic pH is mitigated [124]. Excessive water pollution, especially with high levels of heavy metals, is avoided by snails, and the free-swimming stages of trematodes are also very sensitive to these pollutants [125].

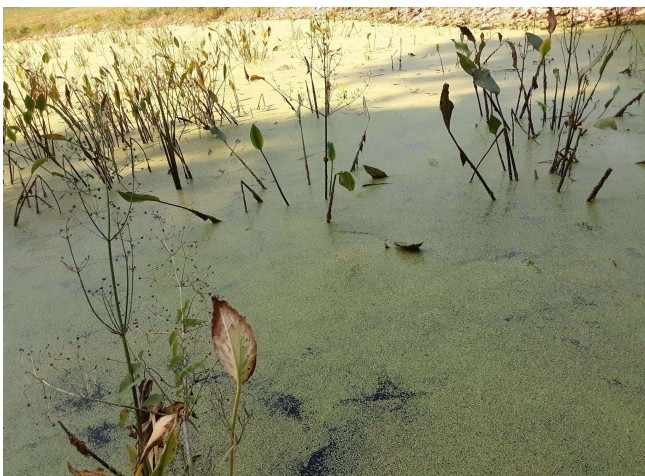

**Figure 8.** Conspicuous signs of eutrophication in a fishpond include diverse swampy vegetation due to the thick silt layer, and the whole surface is covered by duckweeds (Lemnoidae) as a result of agricultural pollution, thus nutrient availability. This type of habitat is appropriate for Lymnaeidae; therefore, agricultural activity can increase the risk of fascioloidosis in European cervid populations.

Water flow velocity has a central role in habitat suitability. Any factor that slows down the water flow in a lotic freshwater habitat can affect the population size of snails. This phenomenon could be observed in Africa after the construction of dams, which expanded the suitable habitat for planorbid snails, increasing the risk of human schistosomiasis caused by *Schistosoma mansoni*, a parasitic trematode [126,127]. The same issue is discussed in Slovakia and Hungary because of the construction of the Gabčikovo-Nagymaros hydroelectric dam, after which *F. magna* occurred within the most affected river section [64]. On the other hand, stormwater runoffs (Figure 9) after heavy rainfalls can remove even half of the macroinvertebrates from the riparian zones of lotic ecosystems [38], which reduces the local number of potential intermediate hosts of trematodes.

Dam constructions do not affect exclusively the water flow but can create artificial barriers in the way of the migration of aquatic fauna, which can also contribute to the invigoration of host-parasite systems. Diama Dam on the Senegal River, after its completion, inhibited the migration of a predator prawn, *Macrobrachium vollenhovennii*. This prawn consumes pond snails; thus, their population decrease resulted in planorbid snail population expansion in prawn-free river sections, which increased the risk of human schistosomiasis in the region [126].

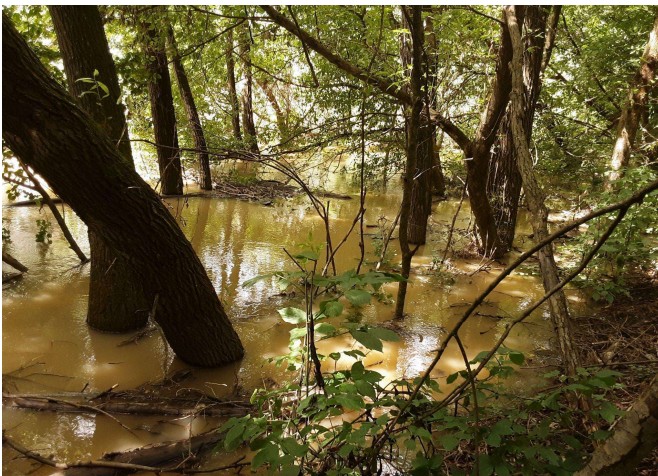

**Figure 9.** Stormwater runoff by the Drava River, a *Fascioloides magna* endemic area. These events can remove even half the snail populations from the riparian zone of the river and contribute to the snail and parasite spreading in the flow direction.

Agricultural cultivation, especially the extensive use of irrigation channels, elevates the nutrition level of freshwater habitats, and climate change amplifies this disadvantageous effect of agriculture [128]. In Europe, this phenomenon can be observed in the Ruhr River water reservoir system. Within the interconnected freshwater ecosystem, the water flow velocity is very low, the water temperature is mild, and sediment accumulation and phytoplankton production are increased, thus eutrophication progressively occurs. In these conditions, a huge *Radix auricularia* population is established, providing a stable environment for a diverse trematode community with both cyprinid and anatid parasites among them [121,122]. In tropical environments, the ambivalent impact of irrigation channels, both the advantage for agriculture and the disadvantage for health, is more conspicuous. Perennial irrigation channel use in Egypt strongly correlates with schistosomiasis incidence in humans [127].

In both lentic and lotic habitats, the floating aquatic plants escalate water flow restriction, thus sediment accumulation, and consequently intensify the eutrophication process (Figure 10) [129]. This phenomenon is widely investigated in African freshwaters, where the water hyacinth (*Eichhornia* syn. *Pontederia crassipes*) is an aquatic ecosystem invader originating from South America. This plant creates dense, mat-forming vegetation, allowing muddy substrate to accumulate around it, which provides food, shelter, and an oviposition site for *Biomphalaria sudanica*, the intermediate host of human schistosomiasis [130,131]. In Europe, *Hydrocharis laevigata*, which is also a South American invader plant, began to spread in non-freezing water habitats. The floating rosettes of the plant block the light, thus fostering anoxic conditions, which contribute to progressive eutrophication and biodiversity loss in native flora and fauna [129]. Due to climate warming, invasive plant species can accelerate the eutrophication process in European aquatic ecosystems [128,132].

*5.5. Human-Mediated Distribution of Epidemiological Risk of Snails*

In the EU, 96 invasive aquatic plant species are recorded, of which 30% have a considerable impact on the ecosystem. The main route through invasive plants entering the European ecosystem is ornamental trade [132]. The most invaded countries in Europe are France, Italy, Germany, and Hungary, with almost 100 invasive plant species in their aquatic ecosystems [128].

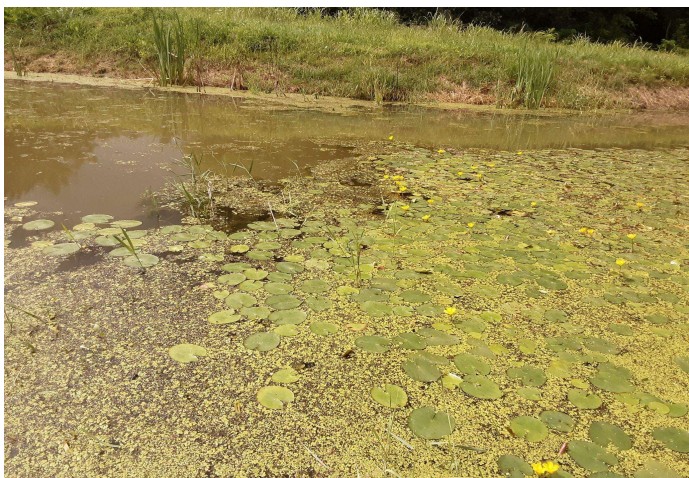

**Figure 10.** Floating aquatic plants, such as *Nymphoides peltata* in the picture, accelerate eutrophication in lotic habitats through flow restriction and sediment accumulation. Some of these plants are invasive outside of their natural range, and global warming can enhance their distribution.

With ornamental aquatic plants, snails are also imported, mostly unintentionally [70], thus the aquarium hobby introduces 41% of the exotic snail species. In most cases, imported exotic snails do not increase epidemiological risk, whereas they have 80% lower parasite richness than native molluscs [133], which can be explained by the Enemy-Release Hypothesis. Hosts can lose their native parasites, establishing their invasive range [116]. However, the imported molluscs can generate significant epidemiological risk [43,134,135].

*Pseudosuccinea columella* is a natural intermediate host of *F. magna* in North America [11,112]. This snail possesses a considerable invasive capacity, whereas it can be found in 22 countries worldwide [42], in various habitat types ranging from tropical climates to high altitudes with very low temperatures [42,134]. Initially, it occurred in botanical gardens in Europe. It is supposed to be imported by ornamental aquatic plants. The first wild population of the species was recorded between 2004 and 2006 in France by the Lot River [136]. By this time, *P. columella* had occurred in different European countries, such as Spain, Portugal, Greece [11], Romania, Italy [42], Austria, Latvia, Czechia [134] and Hungary [134,137].

In France, where the first wild population was detected, *P. columella* aggressively colonised freshwater ecosystems, mainly interconnected water reservoir systems. It can tolerate acidic soil and cool habitats. In competition with native lymnaeid snails, *P. columella* overwhelms both *G. truncatula* and *Omphiscola glabra*. The population decrease of native snails is more rapid in *G. truncatula* than in *O. glabra* [134].

Most populations of *P. columella* are highly susceptible to trematode infections. *Fasciola hepatica* [42] and *F. gigantica* [42,43]; *F. magna* [11,112], and *Fasciola nyanzae* [135] can complete their life cycle utilising *P. columella* as an intermediate host. In an experimental infection with *F. hepatica*, 100% infection prevalence could be reached in *P. columella* [136].

In an Egyptian agricultural environment, irrigation channels are occupied by water hyacinth, which provides an ideal habitat for *P. columella*. As a result, *P. columella* became the most dominant species in that aquatic ecosystem; almost half of the snail individuals belonged to this species. After the local parasite, *F. gigantica*, colonised the invasive species, a spill-back mechanism could be observed. The incidence of *F. gigantica* increased after the settlement of the exotic snail [43]. A very similar phenomenon was noticed in Lake Kariba, where the water hyacinth also provided ideal conditions for *P. columella* invasion. After strengthening the snail population, the hippopotamus population of the lake faced a severe *Fasciola nyanzae* infection, which affected their population dynamics due to the native parasite spillback by the exotic vector [135].

*Pseudosuccinea columella* can extremely quickly adapt to allopatric parasite strains. Pankrác et al. infected Oregon-originating snails with Czech *F. magna* miracidia experi-

mentally. As a result, 74% of the snails became infected [11]. In comparison, experimental infection of *G. truncatula* proved successful in 31–81% [12], while in natural circumstances, the *F. magna* larvae can be found in 0.02–0.03% of the *G. truncatula* [15,97,99]. In allopatric challenge trials, infection of *G. truncatula* with *F. magna* miracidia is less efficient than the encounter of sympatric hosts and parasites [7].

## 6. Conclusions

Advantageous host replacements and environmental adaptation characterised the evolutionary history of the Fasciolidae family. The change in the intermediate host, from the Planorbidae family to Lymnaidae, happened just after the KPgME, which was caused by a climatic catastrophe [1,27,28]. Among those extreme circumstances, the fittest and most abundant potential hosts should be chosen for successful survival. Considering that the Planorbidae family is also a very successful taxon with plenty of species, worldwide distribution [98], and the ability to self-fertilisation, thus population recovery after loss [138], it is supposed that the shift to the Lymnaeidae family is settled locally in those small habitat fragments where the parasite could reach only lymnaeid snails. Though this random host switch proved advantageous.

The high adaptation capacity of *F. magna* to potential intermediate hosts can be realised currently. Dreyfuss et al. investigated a Colombian-origin *Galba* sp. by experimental infection with allopatric, Czech *F. magna*, and French *F. hepatica* through five successive snail generations. They confirmed that during that relatively short period, *F. magna* could adapt to the new host well, whereas both snail survival and cercaria production increased from generation to generation. In the case of *F. hepatica*, this rapid adaptation could not be observed. [139].

The adaptation to highly invasive snail species guarantees the survival and spread of the parasite. The excessive spread of *P. columella* in Europe might open new gates for the further distribution of *F. magna*. This snail can tolerate a wider range of environmental factors than native pond snails, such as *G. truncatula* [134]. Progressive eutrophication, which is enhanced by poor landscape management, e.g., ill-considered construction of irrigation channels [43] or unintentional import of invasive aquatic plants with ecological impact [135], provides suitable habitats for *P. columella*. The ability of invasive species to facilitate each other in their invasive range is investigated in a non-native fish and non-native snail interconnection. The authors ascertained that the common presence of invasive species has a stronger impact on the native ecosystem than if they occurred separately. The non-native elements of the flora and fauna can alter the community structure via alteration of the food web and facilitate each other in the ecosystem. This phenomenon is the invasional meltdown [140].

Exotic aquatic plants can change the flow velocity, the sediment deposition, and the primary production of the ecosystem [129,132], thus changing the living conditions of potential intermediate hosts of trematodes [43]. The extensive agricultural activity as a risk factor for *F. magna* infection is confirmed [73]. The increased nutrient content of water enhances the growth of pondweeds, which provide food and shelter for snails. The slow current in water reservoir systems also threatens by population expansion of snails [121]. The aggressive colonisation of such interconnected aquatic systems by *P. columella* was witnessed in France [134], where the species was detected in a natural ecosystem first in Europe [136]. In its invasive range, *P. columella* successfully displaces *G. truncatula* [134]. The coexistence of *F. magna* and *P. columella* in a natural habitat has not been studied yet.

In the final hosts, Fasciolidae can always be found in the most wide-spread and abundant taxa. During the golden era of the Proboscideans, when this order was the most abundant mammal taxon worldwide, the parasite was supposed to spread by proboscidean radiation [1]. The decline of this host taxon and the rise of ruminants during the last 20 million years promoted the host switch [27]. The opportunistic host selection of the Fasciolidae is conspicuous. Among their hosts, elephants, hippos, other ungulates, and even humans appear. It is hypothesised that this odd mixture of current definitive hosts is

the result of the widespread extinction of the ancient hosts, which forced the parasite to shift to alternative hosts [1].

The large American liver fluke shows very good adaptive capacity. During its evolutionary history, all known host switches proved unimpeded. The jump from proboscideans to Odocoilinae is supported by indirect evidence; however, during the rapidly changing Pliocene, when the megafauna of the Americas decreased, Odocoilinae were the most abundant ungulate taxa [1]. Much later, when the European settlers extirpated the white-tailed deer from the vast majority of its North American territories, the parasite sustained in subpopulations of its native host; however, it adapted to further host species [7].

Smart adaptation to new host populations has been seen recently both in North America and in Europe. Invasion of new territories always causes some negative effects on the host population while the host and the parasite get accustomed to each other. In a North American endemic area, severe clinical signs and a considerable rate of loss could be observed in a freshly infected moose population, while previously infected populations showed mild signs of the infection [93]. In Europe, the populations of large-sized deer species adapted to the parasite some years after its emergence [15]. The most spectacular habituation could be seen in the roe deer species. At the onset of the European invasion, this species was assessed as an aberrant host of *F. magna*, whereas almost all animals died within 6 months after the infection [3,7]. Two decades later, chronically infected roe deer individuals could be detected in two independent investigations in the Drava basin [82,83].

The ability to self-fertilise is also a characteristic of outstanding adaptive capacity. The selfing ability helps species survive population bottlenecks during severe loss or occupation of new distribution areas [83]. Though trematodes are mostly outcrossing animals, self-fertilisation can be detected in 2% of *F. hepatica* [141]. In its invasion range, a parasite must use an alternative strategy to assure the maintenance of the population, even in the absence of a potential mating partner [93]. By selfing, the parasite temporarily gives up the benefits of sexual reproduction in favour of potential expansion. In an investigation carried out within a new distribution area of *F. magna*, the researchers found a 23.5% proportion of selfing in red deer specimens. The eggs originating from selfing did not show significant fitness loss, whereas the embryonic development and hatching ability of inbred eggs were similar to those of outcrossers [83].

In the European distribution area, a pathological variation in wild boar, a dead-end host, could be observed. By investigating 12 wild boars in Croatia, the researchers found a large pseudocyst with a thin wall in one carcass. This pseudocyst formation allows the flukes to survive for a longer time, mature, and produce eggs. In this case, the wild boar, which at present is evaluated to be a dead-end host, might contribute to the maintenance and spread of the parasite in the future [89].

In host-parasite interactions, the two parties continue trench warfare, trying to defeat each other. Meanwhile, as they change each other's population size, the numbers of both hosts and parasites fluctuate over time [94]. In the case of *F. magna*, a considerable population-dynamic effect on the host populations could be seen within the freshly infected areas. The mutual adaptation between the host and the parasite occurred years after the onset of the invasion [10,93]. As if the parasite would cause the absolutely necessary minimal damage to the host population for the sake of subsistence.

From the perspective of final hosts, *F. magna* can be considered an environmentally transmitted parasite that does not depend on host survival. Based on the Curse of the Pharaoh hypothesis, this type of parasite just 'sit-and-wait' in the environment during the population decrease of hosts, as can be observed in the case of *Bacillus anthracis* [142]. However, the environmental survival of *F. magna* is not as long as that of *B. anthracis*. In addition, the real environmental phase of the life cycle is very short. Only the free-swimming miracidia and cercariae live de facto in the environment. The longest stage out of the final host takes place in the intermediate hosts, which must also face the harmful effects of the parasite [3]. Moreover, in the life cycle of *F. magna*, a highly self-fertilising snail taxon [32,37,115,119] meets the mostly outcrosser parasite [83]. By obligate selfing,

the host loses genetic variation, which impedes its adaptation ability during warfare with its parasite. According to the Red Queen hypothesis, this genetic alteration can lead to the extinction of the host population [143].

Despite extreme inbreeding in the Basommatophora taxon, intermediate host species are not threatened by trematode burden. Though invasive populations, which are genetically more monomorphic, are more susceptible to trematode infection than native populations [32,119], this phenomenon does not lead to the collective extinction of snails and their trematode parasites. On the contrary, it increases the epidemiological risk for the final hosts [42,43,120]. Arakelyan et al. investigated a parthenogenetic lizard population, which showed a similar parasite load as the outcrossing population despite its lower genetic variability. The authors concluded that the superiority of sexual reproduction cannot prevail if a large difference exists between the dispersal rates of the hosts and the parasites, and self-fertilising strains possess extra resistance to the parasite [144].

Highly virulent parasites exert considerable influence on the population dynamics of their hosts; therefore, a high prevalence of infection always causes population decline [145]. On the other hand, if the parasite meets a heterogenous host population, significantly less virulent parasites will evolve during the coevolutionary process [146]. In complex communities, where multiple parasites and the presence of predators affect the population dynamics of the host, the accommodation can be more rapid [95]. The parasite must balance between transmission success and virulence, thus reproduction success [142].

In the evolutionary history of *F. magna*, more host population declines and bottlenecks, such as the KPgB climatic catastrophe, proboscidean extinction, white-tailed deer population collapse, and small-scale translocation to Europe, eventuated. None of these host losses are confirmed to be due to the effect of the parasite. However, it survived all these disadvantageous events owing to its opportunistic adaptation to different types of environmental alterations. The large American liver fluke has the ability to utilise new final and intermediate hosts and the determination to adjust its virulence to the host's tolerance. These features are supposed to support the parasite in occupying new habitats as final and intermediate hosts. For efficient forecasting of this process, further research should be necessary with the following aims: to determine potential intermediate hosts (1); to determine if *F. magna* and *P. columella* share any ecosystem in Europe (2); to assess the epidemiological effect of the mutual occurrence of the two non-native species through the invasion meltdown phenomenon (3); to survey the alteration of disease course in different types of hosts, especially wild boars, to assess the epidemiological consequence of a potential change in the pathology of fascioloidosis (4).

**Author Contributions:** Conceptualization, Á.C.; literature review, Á.C., T.H. and G.N.; writing—original draught preparation, Á.C. and T.H.; writing—review and editing, G.N.; visualization, Á.C.; supervision, G.N. All authors have read and agreed to the published version of the manuscript.

**Funding:** This research received no external funding.

**Conflicts of Interest:** The authors declare no conflict of interest.

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
