# Peer review of "The Large American Liver Fluke (Fascioloides magna): A Survivor’s Journey through a Constantly Changing World"

_parasitologia, doi:10.3390/parasitologia3040031_

Round 1

Reviewer 1 Report

 The authors present review of the evolutionary history of F. magna, the distribution by its ancient proboscidean host and the probable drivers of switch to white-tailed deer. They also use the expanding geographical distribution as an example of the introduction and spreading of an animal disease in a new area.

The paper is of interest for the journal as it provides lots of information on a relatively neglected animal disease. For me this paper is acceptable for the journal but needs revision.

Because of the lack of knowledge among veterinarians and parasitologists on this parasite I would recommend to include a chapter on the biology of this parasite, including classification, life cycle (including intermediate host species) and morphology of the different stages. 

Overall, the paper is well written and uses appropriate references. 

Specific comments:

line 17-29:  all organism Latin names should be in italics

Author Response

Dear Reviewer,

Thank you for your supportive suggestions on our manuscript.

As the reviewer recommended, we added a figure to the manuscript with a detailed life cycle and morphological characteristics of different developmental stages of the parasite.

We have not added a whole chapter on biology of the parasite but complemented the Introduction chapter with more details of the life cycle and morphology of F. magna. This solution might be more acceptable for the other reviewer who suggested some shortening of the text.

Thank you for your helpful revision.

Kind regards,

Ágnes Csivincsik

corresponding author

Reviewer 2 Report

Overall it is a detailed review of F. magna evolution. However, a lot of details on early evolution of digenea and the Fasciolidae family are written as fact, however these scenarios are based on limited genomic data, with no fossil records or historic host association studies. It is therefore difficult assess the actual relevance of the review.

At the start of the introduction genus and species of different flukes should be italicized. The first paragraph of the introduction also reads poorly. 

Term final host isnt great, should be definitive host, and the snail hosts should be recognized as intermediate hosts.

Review is fairly dense, with a lot of what seems like possibly irrelevant information. I think it could be shortened substantially and still provide a solid review. 

English is okay, just a few areas where sentence structure is confusing, especially in the first paragraph of the introduction.

Author Response

Dear Reviewer,

Thank you for your work by which our manuscript could get higher quality.

We accepted the reviewer’s suggestion on refining our wording about evolutionary data. In those sentences where the wording suggested absolute confidence, we reworded the text for disambiguation.

We checked the whole text for appropriate use of italic font and corrected the mistakes.

We used the term ‘final host’ exclusively in those cases when we noted all three types of final hosts together. In any other cases, we corrected ‘final host’ to ‘definitive host’, ‘dead-end host’ or ‘aberrant host’.

We accept the opinion that the manuscript is a little bit long. The authors had a lot of discussion on the necessity of some data. This version of the manuscript is the result of a consensus. We agreed that the potential readers do not possess all those information, which help understand the contexture of the review. Therefore, we provided some details on certain facts, which might be boring for some readers, however for others might be very useful.

We hope that this explanation about the lengthiness of our manuscript could be accepted.

Kind regards,

Ágnes Csivincsik

corresponding author

Reviewer 3 Report

Currently, the relevance of research into the settlement of alien organisms is beyond doubt. Invasive species cause colossal economic, environmental, and in some cases social damage, infesting crops, competing with local species, and introducing new types of pathogenic organisms, including helminths, into ecosystems.

At the same time, less attention is paid to studies of the spread of alien species, compared, for example, to alien plant species, including in a number of registers of the most threatened invasive species. In this regard, the authors’ research is relevant and requires support and attention.

There are a number of comments and recommendations for the article that can improve the perception of the material.

1. I believe that it is necessary to provide a distribution diagram (map) of the species, indicating the donor area and distribution routes, natural (where the range includes at least 12 separate isolated areas) and acquired area. There are 8 areas of habitat in Europe and one area in Northern Italy, where it was first noted and described (Bassi, 1875).

Here the species was recorded in Germany (Salomon, 1932) and is actively spreading (Erhardova, 1961) in the Czech Republic and Slovakia (Erhardova-Kotrla, 1971; Rajsky et al., 1994), as well as in the northern part of Hungary (Sztojkov et al., 1995 ), as well as Austria (Pfeiffer, 1983) and Croatia (Marinculić et al., 2002).

Paths indicating the year, detection and type of dispersal (self-dispersal of hosts or removal of infected animals), as well as the type of host - wild, domestic animals will complement the information. Also, I recommend giving a diagram of the life cycle of the helminth in the natural (at least 9 species of animals, as well as domestic animals) and the acquired part of the range (at least 6 species of wild animals, as well as domestic animals), where the range of definitive, dead-end and accidental hosts should be indicated and indicators of helminth infection.

Bassi, R. 1875. Sulla cachessia ittero –verminosa, o marciaia dei cervi, causata dal Distoma magnum. Med. Vet. 4, 497 –515.

Erhardova, B. 1961. Fascioloides magna in Europe. Helminthologia 3, 91 –106.

Erhardova –Kotrla, B. 1971. The occurrence of Fascioloides magna (Bassi, 1875) in Czechoslovakia. Czechoslovak Academy of Science, Prague, Czechoslovakia, pp. 155.

Marinculić, A., N. Džakula, Z. Janicki, Z. Hardy, S. Lučinger, T. Živičnjak. 2002. Appearance of American liver fluke (Fascioloides magna, Bassi, 1875) in Croatia – a case report. Vet. arhiv 72, 319 –325.

Pfeiffer, H. 1983. Fascioloides magna: erster Fund in Österreich. Wien. Tierärztl. Monatschr. 70, 168 –170.

Rajsky, D., A. Patus, K. Bukovjan. 1994. The first finding of Fascioloides magna (Bassi, 1875) in Slovakia. Slov. Vet. Čas. 19, 29 –30.

Salomon, S. 1932. Fascioloides magna, bei deutschem Rotwild. Berl. Tierärztl. Wochenschr. 48, 627 –628.

Sztojkov, V., G. Majoros, K. Kaman. 1995. Szarvasokban ebo nagy amerikai majmetely (Fascioloides magna) megjelenese Magyarorszagon. Mag. Allat. Lap. 50, 157 –159.

2. I propose to point out publications that compare the morphological characteristics of the species in the natural and acquired parts of the range, as well as the relationship of morphology with the host species.

3. Drawings (photos of habitats), although they create a pleasant impression, are large in size and will look better in a diagram, indicating the characteristics of reservoirs (degree of transformation and origin, pH range, flow, shading and other characteristics).

Special notes:

Lines 670–676 “The change in the intermediate host, from Planorbidae family to Lymnaidae, happened just after the KPgME, which was caused by a climatic catastrophe [1, 24–25]. Among those extreme circumstances, the fittest and most abundant potential hosts should be chosen for successful survival. Considering that Planorbidae family is also a very successful taxon with plenty of species, worldwide distribution [84], and ability of selfing, thus population recovery after loss [125], it is supposed that shift to the Lymnaeidae family is settled locally in those small habitat fragments where the parasite could reach only lymnaeid snails." Apparently, the type of intermediate host is not specific, but depends on other environmental factors, as experiments have shown with experimental infection of mollusks.

Line 710: “Proboscideans, cca. 60 –8 MYA, the parasite is supposed to spread by proboscidean radiation [1].” This information was previously presented and in the “Conclusion” section is redundant.

In general, the work requires improving the design, providing generalized data, diagrams indicating data on helminth distribution, infestation, distribution factors and applied control measures - the use of drugs.

Author Response

Dear Reviewer,

Thank you for your plenty of helpful suggestions, which, contributed to the quality of this version of the manuscript.

We added a figure with distribution routes of the parasite. We also accepted the suggestion on further references. Into the Introduction chapter, we added a life cycle diagram with details on biology and morphology on different developmental stages.

About the final hosts, we provided more information on the pathological progress of fascioloidosis. The size of the figures was reduced for better structure of the text.

We added a separate subchapter for control measures.

Thank you for the very constructive revision on our manuscript.

Kind regards,

Ágnes Csivincsik

corresponding author

Round 2

Reviewer 3 Report

I thank the authors for the changes made; the perception of the manuscript has been improved. The main comments were taken into account. Essential information has been added.

Note, format line 341 “finds itself at a dead-end. In horses [86], pigs [87], and cattle, the flukes can mature."

Author Response

Dear Reviewer,

Thank you for your thorough revision. We corrected the mistake.

Best regards,

Ágnes Csivincsik
